# Rapid age-grading and species identification of natural mosquitoes for malaria surveillance

Doreen J. Siria[1,8], Roger Sanou [2,8], Joshua Mitton[3,4,5,8], Emmanuel P. Mwanga [1,3], Abdoulaye Niang [2], Issiaka Sare[2], Paul C. D. Johnson [3], Geraldine M. Foster[6], Adrien M. G. Belem[7], Klaas Wynne [4], Roderick Murray-Smith [5], Heather M. Ferguson [1,3], Mario González-Jiménez [4✉], Simon A. Babayan [3✉], Abdoulaye Diabaté [2,9], Fredros O. Okumu[1,3,9] & Francesco Baldini [3,9✉]

The malaria parasite, which is transmitted by several *Anopheles* mosquito species, requires more time to reach its human-transmissible stage than the average lifespan of mosquito vectors. Monitoring the species-specific age structure of mosquito populations is critical to evaluating the impact of vector control interventions on malaria risk. We present a rapid, cost-effective surveillance method based on deep learning of mid-infrared spectra of mosquito cuticle that simultaneously identifies the species and age class of three main malaria vectors in natural populations. Using spectra from over 40,000 ecologically and genetically diverse *An. gambiae*, *An. arabiensis*, and *An. coluzzii* females, we develop a deep transfer learning model that learns and predicts the age of new wild populations in Tanzania and Burkina Faso with minimal sampling effort. Additionally, the model is able to detect the impact of simulated control interventions on mosquito populations, measured as a shift in their age structures. In the future, we anticipate our method can be applied to other arthropod vector-borne diseases.

[1] Environmental Health & Ecological Sciences Department, Ifakara Health Institute, Off Mlabani Passage, PO Box 53, Ifakara, Tanzania. [2] Institut de Recherche en Sciences de la Santé (IRSS)/Centre Muraz, Bobo-Dioulasso, Burkina Faso. [3] Institute of Biodiversity Animal Health and Comparative Medicine, University of Glasgow, Glasgow G12 8QQ, UK. [4] School of Chemistry, University of Glasgow, Glasgow G12 8QQ, UK. [5] School of Computing Science, University of Glasgow, Glasgow G12 8QQ, UK. [6] Department of Vector Biology, Liverpool School of Tropical Medicine, Liverpool L3 5QA, UK. [7] Université Nazi Boni de Bobo-Dioulasso, Bobo-Dioulasso PO 1091, Burkina Faso. [8] These authors contributed equally: Doreen J. Siria, Roger Sanou, Joshua Mitton. [9] These authors jointly supervised this work: Abdoulaye Diabaté, Fredros O. Okumu, Francesco Baldini. ✉email: Mario.GonzalezJimenez@glasgow.ac.uk; Simon.Babayan@glasgow.ac.uk; Francesco.Baldini@glasgow.ac.uk

Malaria presents a paradox: its transmission depends on mosquito vectors that have a shorter mean lifespan than the malaria parasite requires for its development[1]. Consequently, its persistence depends on the small proportion of mosquitoes that live long enough to transmit malaria sporozoites to a mammalian host. Consequently, small changes in mosquito longevity have a big impact on malaria transmission[2], which explains why malaria control has focussed on interventions that primarily target adult mosquito survival[3]. Examples include insecticidal nets which have substantially reduced the incidence of malaria in Africa[4], but their effectiveness may now be threatened by insecticide resistance[5]. An accurate and reliable assessment of mosquito age structure is crucial for monitoring the impact of vector control interventions. However, current mosquito age-grading methods typically rely on 60-year-old techniques based on ovary dissections[6,7] that are slow, labour-intensive, coarse and imprecise, and which vary between mosquito species[8]. Many alternatives have been investigated with variable success[9–14]. As malaria is transmitted by multiple, often morphologically indistinguishable, mosquito species that differ in longevity, behaviours, and vectorial capacity[15,16], a method that simultaneously estimates vector species and age without relying on time-consuming techniques and expensive reagents would be of great value.

Like all arthropods, mosquitoes have a cuticle whose chemical composition differs between species and changes with age[8]. Infrared spectroscopy can detect changes in mosquito cuticle by quantifying how it absorbs light[13,17]. Early work on infrared spectroscopy for mosquito analysis was restricted to the near-infrared spectrum (10,000– 4000 cm$^{-1}$)[13,17,18].

While near-infrared spectroscopy (NIRS) can distinguish species and age groups with relatively high success in laboratory settings, it has not yet been able to accurately predict the age of mosquitoes in more natural environments[19]. This fall in accuracy is likely due to the greater genetic and ecological variability in wild populations that may affect how the mosquito cuticle develops over time and between populations, even of the same species. Mid-Infrared Spectroscopy (MIRS, 4000–400 cm$^{-1}$) is an alternative technology that, unlike NIRS, measures discrete fundamental vibrations of biomolecules, allowing more information to be extracted from biological samples (such as on protein conformation)[20,21] including detection of more subtle changes among species or mosquitoes of different ages. Recently, we demonstrated that MIRS can accurately predict the species and age structure of African malaria vectors under controlled laboratory conditions[22] but its applicability to ecologically and genetically variable wild mosquitoes is not known yet. In addition, MIRS was used to predict sex, age class (2- or 10-day old), and Wolbachia infection in laboratory-grown colonies of Aedes aegypti mosquitoes[23], as well classifying the species of Aedes aegypti, Ae. albopictus, Ae. japonicus, and Ae. triseriatus[24], both by using Partial Least Squares-Discriminant Analysis on samples collected under laboratory-controlled conditions.

In this study, we aimed to develop a MIRS approach to ultimately predict species and age of natural populations of three major African malaria vectors either raised in semi-field mesocosms or collected from the field. We used a deep-learning MIRS (DL-MIRS) model based on geographically and ecologically distinct female mosquitoes in East and West Africa to reconstruct the age structure of semi-field mosquito populations and detect their changes pre- and post-simulated vector control interventions. Specifically, to predict the age and species of semi-field-reared mosquitoes from a model trained on lab-reared mosquitoes, we used a transfer-learning approach, which took advantage of the convolutional filters learned from training the model on laboratory-reared mosquitoes and retrained the new model on a small sample of independent semi-field-collected mosquitoes,

providing high predictive accuracy on natural mosquitoes. In addition, to test this approach on wild populations, we collected mosquitoes in villages in Burkina Faso and Tanzania, and validated our DL-MIRS model predictions against age structures based on the number of gonotrophic cycles that females underwent, showing high similarity between the two age classification methods. These results demonstrate how this low-cost, artificial intelligence-based approach can determine the age structure of natural vector populations, and constitute a new surveillance tool in the fight against malaria.

## Results and discussion

**Ecologically and genetically variable dataset built for natural mosquito population surveillance.** We created a dataset of mosquito MIR spectra from 41,151 female mosquitoes of three An. gambiae s.l. group species, An. gambiae, An. arabiensis, and An. coluzzii from diverse genetic backgrounds and reared both in different laboratories and in ecologically realistic semi-field systems in East and West Africa to capture laboratory (LV), genetic (GV), and environmental variation (EV) (Fig. 1 and Supplementary Table 1 and Supplementary Table 2). This dataset comprised the LV subset of mosquitoes from three different laboratories in the UK, Tanzania, and Burkina Faso and the GV and EV subsets of adult mosquitoes that were collected from the field as eggs or larvae, or derived from laboratory colonies in Tanzania and Burkina Faso and reared in the laboratory or semi-field systems, respectively (Supplementary Table 3).

First, we used unsupervised clustering of all MIRS data using Uniform Manifold Approximation and Projection v0.5.2 (UMAP)[25]. This revealed signatures within MIR spectra of both geographic origin and rearing environment in all three An. species (Fig. 2a, b). This discrepancy between clusters indicates two properties of these datasets: (i) there exists useful variation between species, suggesting that MIRS are predictive of species, and (ii) there is also unsurprising[19,23] variation between mosquito

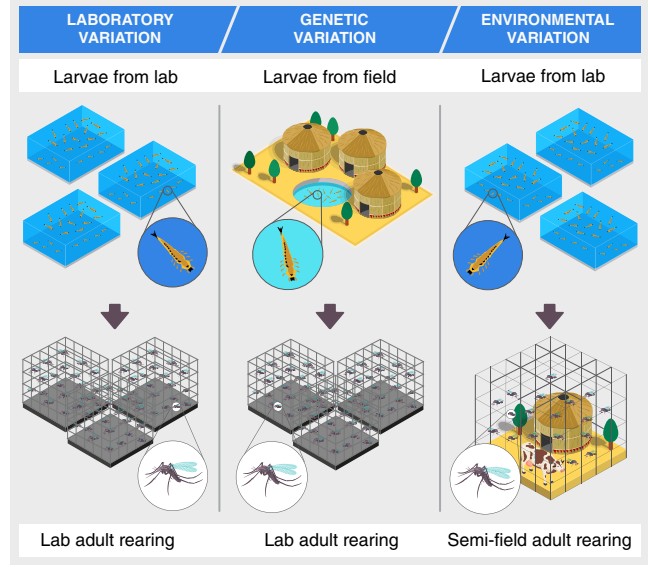

**Fig. 1 Experimental setup for capturing variation in MIRS caused by the laboratory of origin, individual genetic differences and natural environment.** To disentangle genetic and environmental effects, mosquitoes were obtained from either laboratory-bred colonies or from genetically heterogeneous wild larvae; half of the laboratory larvae were then reared and allowed to develop through the adult stage in semi-field conditions, which offer ecologically realistic conditions while still allowing control of mosquito age.

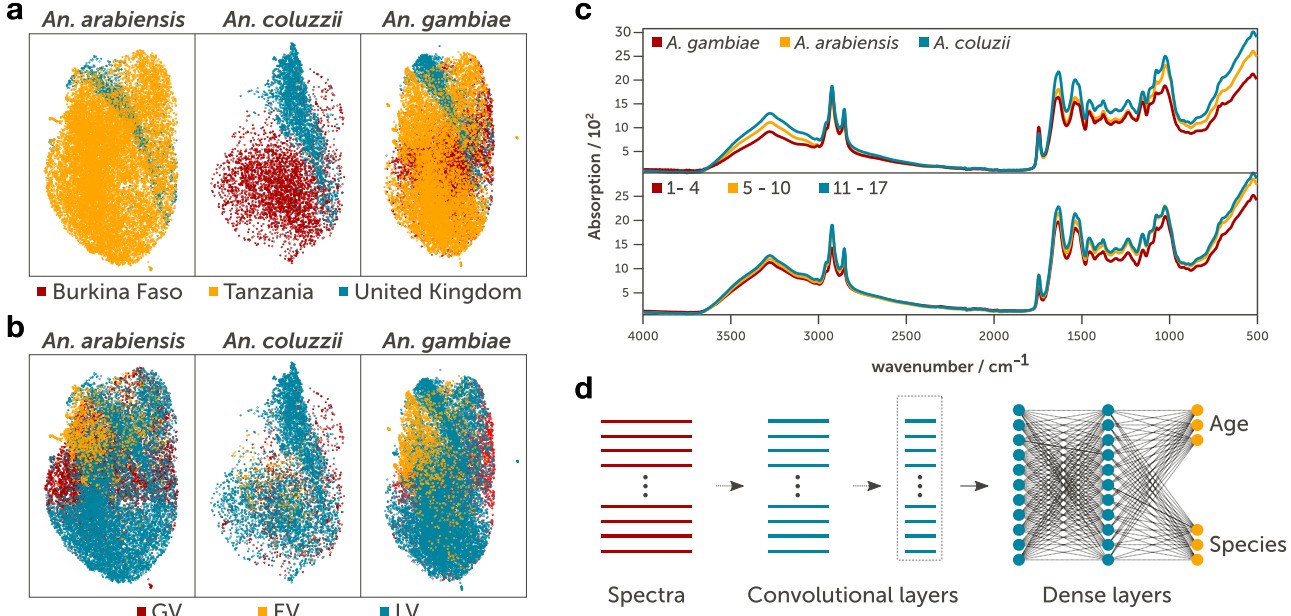

**Fig. 2 Variation in MIRS, machine-learning model architecture, the sensitivity of the trained model.** We collected the MIRS of 41,151 female mosquitoes belonging to three species from diverse laboratories, genetic backgrounds, and environments and three age classes spanning 1-17 days post pupal emergence. **a, b** Unsupervised clustering of MIRS measurements using Uniform Manifold Approximation and Projection of MIRS in two dimensional space (plot axes) from *An. arabiensis*, *An. coluzzii* and *An. gambiae* coloured according to site of origin (**a**) and source of variation (**b**). **c** Representative variation of mid-infrared absorption spectra of *An. arabiensis*, *An. coluzzii* and *An. gambiae* and of three age classes. **d** Schematic representation of the deep convolutional neural network that takes MIRS inputs and outputs mosquito age and species. The input layer (wavenumber values) is fed through five 1-dimensional convolutional layers, comprising of 16 filters each (convolutional layers region), followed by a dense layer of 500 features and age and species output layers (dense layers) that were used to make predictions.

origins and rearing environments, indicating that models based on cuticle composition must include representative samples from each origin to statistically adjust for origin bias.

Mosquito MIRS were then labelled according to their species and one of three age classes (days after pupal emergence) corresponding to their potential to be infected and transmit malaria (Fig. 2c): younger non-infected (1–4 days old), potentially infected but not infectious (5–10 days old) and old enough to be infectious (≥11 days old).

**DL-MIRS transfer learning accurately predicts semi-field-reared mosquito age and species.** The costs associated with rearing and collecting samples from semi-field mosquitoes are substantially higher than for laboratory samples. To minimise end-user investment, we, therefore, adopted a transfer-learning approach to use minimal amounts of semi-field data, while benefiting from a large "one-off" laboratory-reared mosquito dataset. This minimises the computational and data production costs that would be incurred when adapting our models to new wild-living mosquitoes. Specifically, we pre-trained a model on the MIRS from 7200 LV+GV mosquitoes (Fig. 1), which are expected to share many features of wild mosquitoes, then froze the convolutional layers, and re-calibrated the dense layers with MIRS from semi-field mosquitoes using both age class and species as the output layer (Fig. 2d). We thereby retain the feature extraction capability that was learned through training on lab-reared data and utilise this to achieve better predictive accuracy on semi-field data than would be achievable had we purely trained the model on semi-field data (Supplementary Fig. 1). Testing on a separate set of semi-field data of the same geographic origin (Burkina Faso, Tanzania) demonstrates that with a model trained on adult female LV+GV mosquitoes and transfer learning with a further 1452 semi-field (EV) female mosquitoes, DL-MIRS achieved > 95% accuracy in predicting both age (Fig. 3a) and species (Fig. 3b) of unseen mosquitoes reared in ecologically realistic semi-field environmental conditions (EV).

**Transfer learning requires few examples from new target populations.** To assess the smallest number of local samples required by DL-MIRS to perform adequately while minimising field collection effort, we tested model accuracy by using a range of EV mosquito MIRS for transfer learning (Fig. 1d). With no EV samples, the model could predict neither species nor age (Supplementary Fig. 2e, f), regardless of whether we preselected chemically relevant wavenumber values for the input layer as previously described[22] (Supplementary Table 4 and Supplementary Fig. 3). However, increasing the quantity of EV data in the training set caused the prediction accuracy to increase rapidly, already exceeding 80% accuracy with 324 examples and exceeding 90% accuracy when over 815 EV data points were included (Fig. 3c).

To further test the value of the transfer-learning approach, we built models using only the EV included in the training dataset. Without transfer learning, the prediction accuracy was significantly reduced, for example, with 324 and 815 EV training data and no lab data, only 70% and 83% accuracy were achieved, respectively (Supplementary Fig. 1).

In addition, while the model trained on lab-reared data could not predict semi-field samples, on a test set of unseen GV samples it achieved 89% and 95% accuracy on age and species, respectively (Supplementary Fig. 2c, d), highlighting that it had learned features that capture important characteristics of mosquitoes, and which were potentially useful for transfer learning. Further, models trained on only LV mosquitoes achieved 84% and 93% accuracy in classifying age and species of unseen LV data (Supplementary Fig. 2a, b).

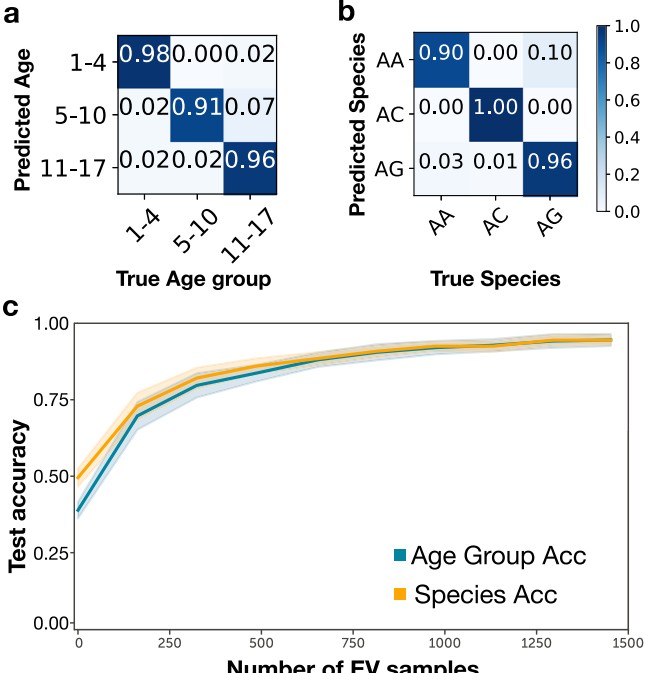

**Fig. 3 Confusion matrices of model prediction accuracies and transfer-learning power.** DL-MIRS was trained using mosquitoes from laboratory larvae reared in the lab (LV, laboratory variation), larvae from the field reared in the lab (GV, genetic variation), and laboratory larvae reared in semi-field (EV, environmental variation). To improve model generalisation from lab to field-reared mosquitoes, we used transfer learning by freezing the convolutional layers of a model trained on LV+GV datasets only and calibrated using a smaller number of EV mosquitoes (here, 1294 examples) to train only the dense layers, resulting in highly accurate identification of (**a**) mosquito age and (**b**) mosquito species. **c** Classification accuracy improved from ~50% to 94% for both age group and species with a training set comprising 0 (i.e. effects of increasing sampling of lab-reared mosquitoes only) through 1452 semi-field (EV) mosquitoes used to re-train the transfer learned model. The solid and shaded lines indicate the mean and standard deviation of the mean of 20 trained models, respectively.

These results demonstrate that transfer learning is an efficient and promising approach for minimising the number of new samples needed for recalibration, making DL-MIRS readily transportable to new mosquito populations.

**DL-MIRS sensitivity to different mosquito cuticle biochemical signatures.** We conducted a sensitivity analysis on the model trained for predicting both LV and GV datasets to understand the regions in the MIR spectra that were the most informative of mosquito age and species (Fig. 4). The sensitivity profiles indicate that the DL-MIRS extracted key biochemical features present in the spectra, corresponding more specifically to wavenumber values associated with the vibration of chitin and protein bonds. Furthermore, the aliphatic hydrocarbon bands (green stripes in Fig. 4) contributed little to the model, suggesting that lipids like wax in the cuticle are less informative in distinguishing age and species of mosquitoes.

**Variation in MIRS profiles between lab and semi-field-reared mosquitoes.** To test whether ecological effects were country-specific, we then trained and tested the DL-MIRS with distinct combinations of mosquito origins. Training a deep convolutional neural network (CNN) including mosquitoes reared in semi-field facilities from one country could not predict age and species of

populations from another (Study E2, Supplementary Fig. 4). Similarly, training a CNN including laboratory-reared mosquitoes from two sites could not predict age and species of those reared at the third (Study E3, Supplementary Fig. 5), even with preselected wavenumber values (Study E4, Supplementary Fig. 3). Further, reducing training input parameters does not improve generalisation, showing that the CNN algorithm is not overfitting (Supplementary Table 4 (Study E1) and Supplementary Fig. 6). These results highlight local variations in MIR spectra even under very similar controlled laboratory conditions and are consistent with the prediction that some spectra from the target population are required to re-calibrate DL-MIRS.

**DL-MIRS detects age structure shifts in simulated mosquito populations following vector control interventions.** Next, we evaluated how well DL-MIRS could detect the impacts of vector control on simulated wild mosquito populations. We have previously demonstrated that models trained on MIRS from laboratory-reared mosquitoes can be utilised to reconstruct the age structure of simulated populations from which they were sampled, and correctly detect whether those populations had been subjected to long-lasting insecticide-treated nets (LLINs)[22]. However, it remains unclear if our present DL-MIRS model can achieve similar performance on genetically- and ecologically-diverse mosquitoes. We therefore investigated the power of DL-MIRS to detect changes in the age structure of simulated mosquito populations subjected to two vector control interventions selected to reflect the probable impacts of current and next-generation vector control strategies: (i) intervention with a rapid killing effect as expected from long-lasting insecticide-treated nets (LLIN); (ii) intervention with the slower killing effect that primarily impacts "old" mosquitoes as expected with attractive toxic sugar baits (ATSB) (Fig. 5a).

We first simulated the age structure of mosquito populations before and after LLIN or ATSB intervention estimating 9% mortality for the control group from previous reports[26], and assuming 36% mortality for mosquitoes above three days old for LLIN or constant 18% mortality for ATSB. Then, we estimated the statistical power of DL-MIRS to detect shifts in the three age groups (non-infected, potentially infected, potentially infectious) anticipated from these interventions (Fig. 5b, c, dotted line). We then assessed how the power to detect age structure shifts varied with the number of local mosquitoes used for model testing (Supplementary Table 5; see 'Methods' for power analysis details). In both intervention scenarios, sampling 300 mosquitoes pre- and post-intervention was sufficient to obtain >80% power to detect an age structure shift when the training set was composed of 162 EV mosquito spectra (Fig. 5b, c, solid lines and Supplementary Fig. 7). This shows that with relatively minor sampling and machine-learning training efforts, this approach is capable of detecting population age structure shifts following vector control interventions.

**DL-MIRS predicts wild mosquito physiological age.** Next, we evaluated the ability of DL-MIRS to predict the age of wild mosquito populations. As no measure of chronological age exists for wild mosquitoes, we used the Polovodova ageing technique[27] as an independent gold standard for estimating age, as defined by the number of gonotrophic cycles females mosquitoes have completed at the time of capture. The number of gonotrophic cycles is assessed through observation of ovarian morphology. Mosquitoes were collected in Tanzania and Burkina Faso from villages where only one *An. gambiae* s.l. species was expected. This was confirmed through PCR identification of a subset of collected mosquitoes in each village, *An. coluzzii* in Burkina Faso and *An. arabiensis* in

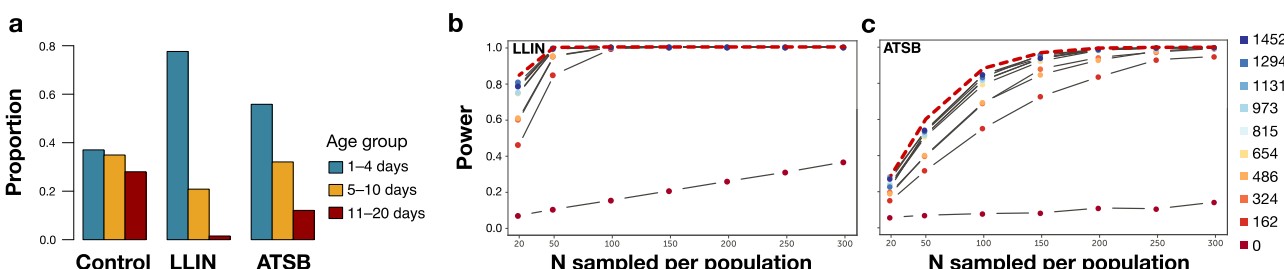

**Fig. 4 Average model sensitivities to different wavenumber values and comparison with the features of the average absorption spectrum (grey line) of each output class.** The coloured stripes show the regions associated with the particular vibration of a functional chemical group. The upper part (maxima) displays the intervals of wavenumber values in which the maximum of the absorption peaks of each vibration appear for each of the three most abundant components in the cuticle of a mosquito[22]. Here, the vibration of the same bonds appears in different wavenumber values depending on which cuticular component they belong to (chitin, protein or wax), which modifies the shape of the peaks.

**Fig. 5 DL-MIRS generalisation and detection of vector control intervention. a** Computer simulations were used to assess the power of DL-MIRS a 'rapid kill' (long-lasting insecticide-treated nets; LLIN) or 'slower kill' intervention (attractive toxic sugar baits; ATSB) relative to a population with no intervention (control). **b, c** Power to detect an effect of the vector control intervention was estimated over 10 levels of training set size represented by coloured points, with EV mosquitoes ranging from 0 to 1452 and seven sample sizes per population from 20 to 300 (Supplementary Table 5). The dashed red line shows the power that would be achieved with 100% accurate age group classification. The difference between the solid and dashed lines represents the cost in power due to prediction error.

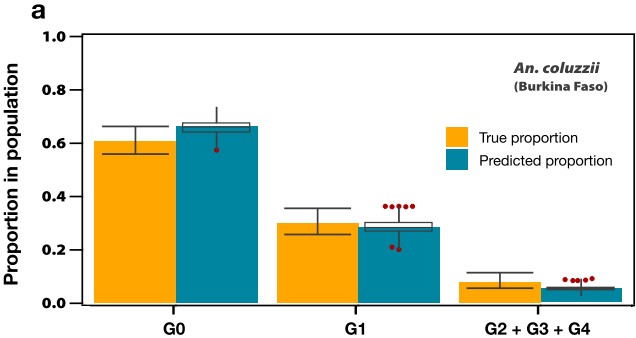
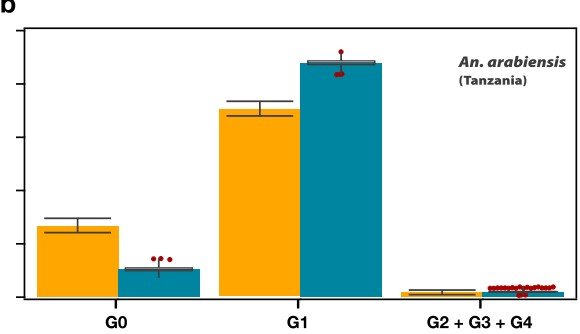

**Fig. 6 DL-MIRS validation on wild mosquito populations.** The proportion of wild female mosquitoes with 0, 1 or ≥2 gonotrophic cycles (G0, G1, G2+G3+G4) was determined by ovarian dissection and morphological characterisation (yellow) or predicted by DL-MIRS on non-dissected mosquitoes (blue). The same number of mosquitoes for each group was analysed on each day of collection, either from Burkina Faso (**a**, *An. coluzzii*) or from Tanzania (**b**, *An. arabiensis*). The mean proportions and 95% credible intervals of the age proportion from dissected mosquitoes (yellow) were estimated with a Dirichlet distribution. The age proportion predicted by the DL-MIRS (blue) is presented as box-whisker plots showing the median, interquartile range (IQR, box), lowest/highest data within 1.5 IQR (whiskers), and outliers (red points) of the probability distribution of predictions from ten different models.

Tanzania (Supplementary Table 2). Live collected mosquitoes were killed and dissected either on the same day, or 2–3 days after field collection to allow for oviposition in cages. The number of gono-trophic cycles passed was morphologically identified (Supplementary Table 6). After dissection, mosquitoes were dried and scanned by MIRS. Gonotrophic cycle classification was used to (i) estimate the overall physiological age structure over the collection period at each site, and (ii) provide known age classes for transfer learning of DL-MIRS spectra. To independently test model predictions, we also scanned non-dissected mosquitoes, selected at random from the same populations as the dissected ones. Here, we assumed that the age structure of dissected and non-dissected mosquitoes should be similar. We retained the convolutional layers of the previous model trained on 7200 LV+GV mosquitoes, and retrained the dense layers with MIRS from the wild dissected mosquitoes using the number gonotrophic cycle as the true output classification. Although the convolutional layers we used were built to predict chronological age classes, we expect these to share many features of wild mosquitoes classified into gonotrophic cycles. Indeed, the three classes of 1–4, 5–10 and ≥11 days old correspond to females that underwent 0, 1 or ≥2 gonotrophic cycles[28]. Separate models were trained with 335 wild mosquitoes collected in Burkina Faso and 758 from Tanzania. DL-MIRS predicted very similar age structures for non-dissected (test) and dissected (morphologically assessed) wild mosquitoes (Fig. 6). This suggests that DL-MIRS can be readily adapted to diverse field settings and ageing methodologies.

Malaria continues to be a major cause of mortality and economic hardship in communities across the world. Vector control remains a primary weapon against it and has generated substantial progress in reducing malaria burden worldwide. However, the epidemiological impact of such interventions can be difficult and time-consuming to assess, requiring laborious and costly large-scale trials. Here, we have demonstrated that an approach based on transfer learning has the potential to overcome this limitation, and facilitate prediction of age structure in diverse malaria vector populations with limited sampling. Building on an earlier proof-of-principle that MIRS and machine learning allow high throughput speciation and age classification in laboratory settings[22], here we demonstrate that deep-learning models pre-trained on cheap, large-scale MIRS datasets from laboratory-reared mosquitoes could be rapidly transferred to new ecologically realistic mosquito populations with only a few mosquitoes (<1000) from the target population. Further, the impact of vector control interventions such as LLIN and ATSB could be detected by a DL-MIRS model re-calibrated on those populations with >90% accuracy with as few as 150 mosquitoes

sampled before and after the intervention. Finally, validation of this approach on wild mosquito populations whose physiological age was inferred by ovarian morphological characterisation shows that DL-MIRS is an approach that can learn and predict physiological age structure in the field. Indeed, while estimates of physiological age based on gonotrophic cycles are prone to errors (for example, in counting hard-to-see follicular sacs in ovarioles), potential limitations (e.g. due to the uncertainty of classifying blood-fed and gravid mosquitoes into a specific age class), and variation across sites, the overall concordance between the predicted and observed age estimates across two countries suggests that DL-MIRS can accurately predict the age structure of wild populations.

In the future, the accuracy of DL-MIRS for mosquito analyses could be substantially enhanced through the enrichment of training sets with more spectra from additional populations and colonies. Furthermore, the use of additional age-grading methods for wild mosquitoes (for example measuring sporozoite rates) would be crucial to further validate the DL-MIRS approach on field populations. Currently, while implementation of this approach in the field requires relatively minor calibration of the target population, we envision that the inclusion of environ-mental data known to influence ageing rate, such as temperature, is likely to further triangulate predictions and increase prediction accuracy and generalisability, further reducing the need for local calibration. This approach would require an initial investment for the ATR-FTIR spectrometer (~$20,000), but no other costs thereafter. Consequently, DL-MIRS holds great promise and potential for integration into vector surveillance, where it could play a key role in enhancing control and winning the fight against mosquito-borne diseases such as malaria.

## Methods

**Study sites.** The experimental studies were conducted in two leading African malaria vector control institutions: Ifakara Health Institute (IHI), Tanzania and Institut de Recherche en Sciences de la Santé (IRSS), Burkina Faso. In IHI, the study experiments were conducted in the mosquito biology laboratory Vector Sphere and the larvae were collected from three villages in the Kilombero flood-plains in Ulanga district, south-eastern Tanzania: Minepa village (8.285°S, 36.669°E), Tulizamoyo village (8.348°S, 36.732°E), and Sululu village (8.003°S, 36.832°E). The ecology and species available in Ulanga district were recently described[29]. In IRSS, mosquito sampling was conducted in the north of Bobo-Dioulasso in Vallée du Kou village VK5 village (4.4201°W, 11.3824°N) and in the south of Bobo-Dioulasso in Soumousso (4.0438°W, 11.0125°N).

**Mosquito collection and rearing.** We collected *An. arabiensis*, *An. gambiae* and *An. coluzzii* mosquitoes born either from lab colonies or from wild mosquitoes, and reared either in the laboratory (at University of Glasgow [UoG], IHI or IRSS)

or semi-field environments (at IHI or IRSS) (Fig. 1). Laboratory, specific semi-field and wild validation methods are described below.

*Laboratory colonies.* Mosquitoes were reared in the three different insectaries maintained under controlled temperature and humidity and a 12 h:12 h (light:dark) photoperiod, following standard operations[30]. Adult mosquitoes were fed with 5–10% sugar solution ab libitum via filter paper. Mosquitoes were provided with blood meals to allow egg production. Blood meals were provided using human blood at IHI directly by a human arm and at UoG through membrane feeding following[31], and rabbit blood at IRSS directly on the animal. In each institution, different malaria vector species and strains were reared, as indicated in Supplementary Table 1. To produce age-matched mosquitoes, pupae were added to a separate empty cage on the same day. To generate different reproductive conditions, they were blood-fed on different days after emergence and allowed to lay eggs in an oviposition cup 2 days after each blood meal. Mosquitoes were collected either 2 days after a blood meal (i.e., before egg laying) or 4 days after the blood meal (i.e., after egg-laying had occurred). Mosquitoes were starved for 6–12 h by removing sugar before blood-feeding, and each cage was blood-fed every 6 days. Thus, mosquitoes living 6 or more days after their first blood meal underwent multiple gonotrophic cycles. Mosquitoes were sampled at ages ranging from 1 to 17 days old. A hundred and twenty mosquitoes per day (age) were assessed, comprising each of the three physiological statuses.

*Field-collected mosquitoes reared in the laboratory.* At the Ifakara Health Institute, *An. arabiensis* larvae at different stages were collected from Minepa and Tuliza-moyo villages at different aquatic habitats. Larvae were brought to the insectary and were sorted based on their morphology. The larvae were maintained in field water and provided with ground fish food (TetraMin®) until pupation. A plastic pipette was used to transfer pupae from the basins into disposable cups, which were then placed inside $30 \times 30 \times 30$ cm cages until they emerged as adults. Emerged female and male adults were kept together to allow mating. These adult field-derived mosquitoes were maintained at a same temperature $27 \pm 1.0$ °C, humidity $80 \pm 5\%$ and a 12 h:12 h (light:dark) photoperiod, as lab-reared mosquitoes as previously described. At IRSS, all female mosquitoes, whether blood-fed or gravid, were collected by trained technicians with mouth aspirators from local houses where mosquitoes have rested after a blood-feeding in VK5 and Soumousso villages. After aspiration, mosquitoes were transferred immediately into $30 \times 30 \times 30$ cm cages covered with a wet cloth to avoid dehydration during transport. These mosquitoes were transferred to a room where light, humidity, and temperature are similar to that of the field (semi-field facility) and were maintained with glucose 5% for 72 h to allow them to digest the blood. Then individual gravid females were transferred to a single cup containing 10ml water to allow for oviposition. After 2 days, the females that laid eggs were removed with forceps and fixed in 80% ethanol for molecular species identification as previously described[32]. After oviposition, only *An. gambiae* and *An. coluzzii* offspring from Soumousso and Vallée du Kou were kept and reared until adulthood for MIRS sample collection. At both sites, blood-feeding and oviposition occurred in the same way and with the same timings as described for laboratory mosquitoes and a hundred and twenty mosquitoes per age (1 to 17 days) were assessed.

*Laboratory colonies reared in the semi-field.* At each site, experiments were conducted during the rainy season, which is when mosquito populations peak. Temperature and humidity were monitored every day in the semi-field and recorded. In addition, mosquitoes had access to live cattle for blood-feeding and the semi-field chamber included water containers for mosquitoes to lay eggs in. These water containers were checked each day when adult collections were performed. The water was discarded and replaced daily in order to prevent the emergence of new adults. This ensured a single age group was present in each semi-field enclosure. Pupae from mosquito colonies (Supplementary Table 1) were released into the semi-field on two consecutive days and then recaptured at specific days for MIRS sampling. Day 0 was considered the day after the last batch of pupae was released into the facility. At IHI, mosquitoes were collected from day 1 to 17 in batches of 100 mosquitoes per species and per age, while at IRSS mosquitoes were collected on days 1, 4, 7, 10 and 15 in batches of 50 mosquitoes per species and per age.

*Validation using field-collected wild mosquitoes.* At IHI, female mosquitoes were collected using human baited double net traps[33] from 6 pm to 6 am in Sululu village between May and August 2021. At IRSS, female mosquitoes were collected inside houses using mouth aspirators from 6 pm to 7:30 am in VK5 between May and July 2021. At both sites, immediately upon collection mosquitoes were transferred into a cage and provided with 5% glucose solution, covered with a wet cloth to avoid dehydration, brought to the lab, and kept under standard conditions as described earlier. Mosquitoes were then dissected on the same day or 2–3 days after collection to determine the number of gonotrophic cycles that they underwent, using the Polovodova technique[27]; briefly, the head/thorax was separated from the abdomen and preserved in silica for subsequent MIRS measurement. The ovaries were then removed from the abdomen and individual ovarioles inspected for the number of follicular sacs, corresponding to the number of completed gonotrophic cycles. As the gonotrophic cycles can be determined neither in mosquitoes that have blood in the midgut, nor in those that are undergoing egg

development, female that could visually be determined as gravid or blood-fed were not dissected on the day that were collected, but left in the cage for 2 or 3 days with an oviposition cup to allow for blood digestion and oviposition. However, some mosquitoes were still found to contain developing oocytes or eggs at the time of dissection (gravid); these were assigned to a gonotrophic cycle as described below, but their mid-infrared spectra were not included in model retraining. Each day that mosquitoes were dissected, an equal number of non-dissected mosquitoes was killed with chloroform and kept in silica for subsequent MIRS measurement from the same batch (i.e. mosquitoes collected from the same village on the same day); this non-dissected group was used as the unseen dataset to be predicted by DL-MIRS. In each village, a subset of mosquitoes was confirmed to be *An. arabiensis* (IHI) or *An. coluzzii* (IRSS) by PCR, confirming previous findings that these species are the most dominant *An. gambiae* s.l species in these villages[29,34].

**Spectroscopy.** Upon collection of mosquitoes for all experiments conducted either in the laboratory, field or semi-field, mosquitoes were firstly transferred into a cup and then killed with cotton-soaked chloroform[22]. Afterwards, mosquitoes were dried in a tube with silica gel desiccant for at least three days and then measured with MIRS[22]. A single IR spectrum of each mosquito was acquired using Bruker Vertex 70 (UK) and Bruker ALPHA (Burkina Faso and Tanzania) FTIR spectro-photometers equipped with a diamond ATR accessory using Bruker OPUS Software. To maximise the contact of the mosquito cuticle with the ATR crystal, the samples were pressed against the crystal using the anvil attached to the instrument. Background and MIR spectra were acquired by averaging >16 scans at a resolution of 4 cm$^{-1}$ over a range of 500–4000 cm$^{-1}$. Mosquito spectra with low intensity or a significant atmospheric intrusion were discarded automatically using a custom script[22,35].

**Laboratory and semi-field mosquito MIRS datasets.** The core datasets used are built from samples collected as described in Supplementary Table 2, allocated between training and test sets as detailed in Supplementary Table 3, and modified for studies provided in the supplementary text as detailed in Table 4. All datasets were used for the prediction of mosquito age and species, with age labelled as a categorical outcome with three levels of ages 1–4 days, 5–10 days, and 11+ days. Species was also a categorical with three levels, *An. arabiensis*, *An. gambiae* and *An. coluzzii*.

**Machine learning: building DL-MIRS.** We trained a deep convolutional neural network (CNN) using 1D convolutional layers to predict both age and species from MIRS input data. We chose CNNs because mid-infrared spectra are composed of multiple peaks which capture the biochemical characterisation of mosquitoes; a network that includes numerous convolutional layers is capable of capturing complex local features in the spectra. In addition, fully connected layers can combine features learned by the convolutional layers to capture the correlation of features across the entire spectra, a necessity for the high-dimensional spectra used for analysing mosquito cuticles. Each 1D convolutional layer defines filters of fixed width with trainable weights and biases, where each convolutional operation is the sum of the dot product between the filter and the section of the spectra currently considered. The assumption of locality made in convolutional neural networks holds when using mid-infrared spectra due to fixed-width wavenumber bands corresponding to individual vibrational modes. Consequently, convolutional layers were able to learn local structure in the spectra. In addition, we applied batch normalisation to improve the stability of the neural network and max pooling was used to reduce the spatial size of the representation. Further, $L_2$ regularisation was used in each layer to reduce overfitting, with a convolutional stride of size one in convolutional layers one, three and five, and stride of two in convolutional layers two and three. We used size two max pooling in the final convolutional layer, and dropout was applied before the dense layer to further reduced overfitting. The convolutional neural network architecture was found by optimising the hyper-parameters. For this, the number of layers was hand optimised, while the kernel, stride, and pooling sizes for each convolutional layer were optimised using gp_minimize from scikit-optimise v0.8.1[36].

Unless otherwise stated, we trained the DL-MIRS CNN using datasets balanced across mosquito age groups and species. We began by splitting out 10% of the dataset stratified by age and species for subsequent testing of the trained models. The remaining 90% of the dataset was used for optimising models through 10-fold cross-validation. Both age and species groups were binarised using MultiLabelBinarizer[36] and the spectra were standardised using StandardScaler[36] to centre each variable around its global mean and scale it to unit variance. Machine learning was performed in Python v3.6.8[37] using keras v2.2.4[38] and tensorflow-gpu v1.12.0[39].

When DL-MIRS was used on semi-field or wild specimens, the model was not trained in its entirety and instead transfer learning was used. For this, we froze the weights in the convolutional layers and the model only updated the weights in the dense layers. This ensured that the model retained the same convolutional features that were previously found. In both cases of retraining, we used the model previously trained on lab-reared data (LV+GV). When retraining the models on wild data (for which age was based on ovarian characterisation and gravid

mosquitoes were excluded), the datasets were imbalanced (Supplementary Table 3), so the optimisation gradients were re-weighted to mimic a balanced dataset.

Instead of taking the max value of the output probability distribution to be the predicted class as in the rest of our results, for the wild data we took the predicted probability distributions of the model since we were comparing the distribution of dissected mosquitoes to that predicted by the model. For this, we made a probabilistic prediction for each mosquito over the three classes. We then sampled from this distribution 100 times and then took the mean over each of the mosquito data points, yielding 100 samples for each model. We present these averaged predictions of the probability distribution for the wild mosquito results.

**Power estimation of vector control detection**. To estimate the statistical power of the DL-MIRS model to detect a shift in the age structure of a mosquito population after each of two common insecticidal interventions, we generated computer-simulated mosquito populations exposed to two different intervention classes that vary in the expected speed of killing effect: (i) fast killing effect as represented by long-lasting insecticide-treated nets (LLIN), and (ii) slower acting killing effect as represented by attractive toxic sugar baits (ATSB). The age structure (i.e., the frequency of each age class) in the control (pre-intervention) population was simulated assuming a constant daily mortality of 9%[26] up to 20 days, with no survival after 20 days post-emergence. The LLIN intervention was assumed to cause a death rate of 36%, four times higher than natural mortality, but applying only after day 3 assuming mosquitoes will host seek and encounter a bed-net only after this age. In the population exposed to the ATSB intervention, mortality was assumed to be two times higher (18%) than natural mortality, applying throughout the mosquitoes' lives. The age structure of each post-intervention population was then compared with the control population using Wilcoxon/Mann–Whitney $U$ tests. Power was estimated across all 70 combinations of seven sample sizes ($n = 20, 50, 100, 150, 200, 250$ and $300$) and ten levels of enrichment of the training data with EV data (0–17% of the training data was EV data). For each of these 70 scenarios, power was estimated as the proportion of 10,000 simulated datasets where a significant ($P < 0.05$) difference in age structure was detected between intervention and control populations. Simulations were performed in R v3.6.1[40].

**Modelling the age structure from gonotrophic cycles**. The posterior distribution of the proportion of female mosquitoes in each age class (0, 1 or ≥2 gonotrophic cycles), $P = (p_0, p_1, p_{\geq 2})$, was modelled as a Dirichlet distribution,

$$P \sim \text{Dirichlet}(N_0 + \alpha, N_1 + \alpha, N_{\geq 2} + \alpha) \tag{1}$$

from which mean proportions and 95% credible intervals were calculated. $N_0$, $N_1$ and $N_{\geq 2}$ are the observed counts in each class and $\alpha = 1$ is the concentration parameter of the prior Dirichlet distribution.

**Ethical statement**. This study has been agreed by the institutional ethical committee of Institut de Recherche en Sciences de la Santé (IRSS) under the number A012-2017/CEIRES on July 3, 2017 before its implantation on the sites. At Ifakara Health Institute, Ethical approval for the study was obtained from the Ifakara Health Institute Institutional Review Board (Ref. IHI/IRB/EXT/No: 005-2018), and from the Medical Research Coordinating Committee (MRCC) at the National Institutes of Medical Research (NIMR), Ref: NIMR/HQ/R.8c/Vol.II/880. At the University of Glasgow, human blood for feeding female mosquitoes was obtained from the Glasgow and West of Scotland Blood Transfusion Service. Ethical approval for the supply and use of human blood was obtained from Scottish National Blood Transfusion Service committee for the governance of blood and tissue samples for non-therapeutic use, and Donor Research (submission Reference No 18 15). Whole blood from donors of any blood group was provided in Citrate–Phosphate–Dextrose–Adenine (CPD-A) anticoagulant/preservative. Fresh blood was obtained on a weekly basis.

**Reporting summary**. Further information on research design is available in the Nature Research Reporting Summary linked to this article.

## Data availability

The mid-infrared spectral data generated in this study have been deposited in the Enlighten database and are available at https://doi.org/10.5525/gla.researchdata.1235. All other data generated in this study are provided in the Supplementary Information/Source Data file. Source data are provided with this paper.

## Code availability

All code used for machine learning and power analysis is available at https://github.com/SimonAB/DL-MIRS_Siria_et_al[41].

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

## Acknowledgements

This work was funded by the Medical Research Council GCRF Infections Foundation Awards MR/P025501/1 to A.D., F.B., F.O.O., H.M.F. and K.W. A.D., F.B., F.O. and S.A.B. were supported by the Royal Society International Collaboration Award ICA/R1/191238 and Bill and Melinda Gates Foundation award OPP1217647. F.O. was also supported by a Wellcome Trust Intermediate Fellowship in Public Health and Tropical Medicine (Grant Number: WT102350/Z/13), F.B. by an AXA RF fellowship (14-AXA-PDOC-130) and an EMBO LT fellowship (43-2014). K.W. and M.G.J. thank the Engineering and Physical Sciences Research Council (EPSRC) for support through grants EP/K034995/1, EP/N508792/1, and EP/N007417/1, the Leverhulme Trust through Research Project Grant RPG-2018-350, and the European Research Council (ERC) under the European Union's Horizon 2020 research and innovation program (grant agreement No. 832703). J.M. is supported by a University of Glasgow Lord Kelvin Adam Smith Studentship. R.M.-S. is grateful for EPSRC support through grants EP/R018634/1 and EP/T00097X/1. We would like to thank Dorothy Armstrong and Elizabeth Peat for their assistance with mosquito rearing and maintenance. We would also like to thank Hilary Ranson for providing the Kisumu colony. We thank Fibios Science Communication (Graph Your Science) for help with Fig. 1.

## Author contributions

Conceptualisation: A.D., D.J.S., F.B., F.O.O., H.M.F., J.M., K.W., M.G.-J., R.M.-S., R.S. and S.A.B. Data curation: A.N., D.J.S., E.P.M., F.B., I.S., J.M., M.G.-J. and R.S. Formal analysis: J.M., M.G.-J., P.C.D.J., R.M.-S. and S.A.B. Funding acquisition: A.D., F.B., F.O.O,. H.M.F., K.W. and R.M.-S. Investigation: A.N., D.J.S., E.P.M., F.B., I.S., M.G.-J. and R.S. Methodology: A.D., A.N., D.J.S., E.P.M., F.B., F.O.O., G.F., H.M.F., J.M., K.W., M.G.-J., P.C.D.J., R.S. and S.A.B. Project administration: A.D., A.N., D.J.S., F.B., F.O.O., H.M.F. and R.S. Resources: A.D., A.M.G.B., F.B., F.O.O., H.M.F. and K.W. Software: J.M., M.G.-J., R.M.-S. and S.A.B. Supervision: A.M.G.B., A.D., F.B., F.O.O., H.M.F., K.W., M.G.-J., R.M.-J. and S.A.B. Validation: F.B., J.M., M.G.-J., M.G.-J., P.C.D.J., R.M.-S. and S.A.B. Visualisation: D.J.S., F.B., J.M., M.G.-J., P.C.D.J., R.S. and S.A.B. Writing—original draft: D.J.S., F.B., J.M., M.G.-J., P.C.D.J., R.S. and S.A.B. Writing—review & editing: all authors.

## Competing interests

The authors declare no competing interests.
