## [Peer Review File · Nature Communications]

Rapid age-grading and species identification of natural mosquitoes for malaria surveillanceReviewers' Comments:

Reviewer #1:

Remarks to the Author:

New methodology for the accurate and rapid age classification of mosquito disease vectors would be an enormous asset to the evaluation of anti-vector control measures and the field of vector biology as a whole. This paper "Rapid ageing and species identification of natural mosquitoes for malaria surveillance" by Siria et al. is an effort continuing their 2019 Wellcome Open Research manuscript on the use of mid-infrared spectroscopy (MIRS) to age/speciate *Anopheles* spp. mosquitoes. I commend the authors for their work in generating a very large dataset (almost certainly the largest of its kind), with lab/semi-field samples, multiple countries, species, and mosquito physiological statuses.

The paper is well-written and scientifically sound description of their current efforts, though I worry that the tone is overly positive regarding the data as presented. A critical issue with past work in this field using the related technique of Near Infrared Spectroscopy (NIRS) is that while you achieve good accuracy for laboratory colony mosquitoes, when you make the jump to field-caught material you essentially lose predictive power. This is likely due to a variety of genetic and environmental factors, which happily, the authors have attempted to account for here through their study design and the inclusion of mosquitoes from a variety of labs and rearing conditions. Unfortunately much like NIRS, it appears that MIRS may have similar issues with generalizability of the approach. The authors demonstrate that you must include some portion of mosquitoes from the test group (here 17%--- though with large sample sizes), for predictive ability to be achieved. I think that on its surface this necessary inclusion may be the "tax" necessary to use these techniques, but I feel that the authors have insufficiently demonstrated what this may look like in real world practice and the eventual move to field mosquitoes.

The study brings wild-caught larvae into semi-field environments to improve predictive ability, but it is not demonstrated that this inclusion gives you sufficient accuracy on new, independently reared groups of semi-field mosquitoes, nor (ideally) a selection of fully wild adult mosquitoes. While fully wild mosquitoes captured as adults would have unknown ages, proxies of age such as nulliparity (probably young), sporozoite positivity (certainly old), or even the general predicted age distribution of the population (skewed young) would be good indicators that the methodology is working. I acknowledge that analysis of full field adults is difficult, though this group certainly has the skills and ability to do so.

If field mosquitoes are not available, I believe that at a minimum a demonstration of the accuracy of the technique on these different cohorts of semi-field reared mosquitoes would also be a sufficient demonstration that the technique is not overfitting and would have generalizable success. I think that transparency in these results is critical to pushing this avenue of research forward, and to not have other researchers purchase this instrument/putting time into this for wild mosquitoes erroneously believing it would give them success.

Specific suggestions (some themes in the comments may be repeated, so feel free to ignore if previously answered):

Line 30: Specific what this 95% accuracy refers to, species/age grading/both?---comparing what model set to what test set?

Line 33-34: Final sentence of the abstract is awkward, maybe "In the future, we anticipate our method can be applied..."

Line 50, insert comma after "indistinguishable".

Line 82-86: are these "natural mosquito populations" or would it be more accurate to call them semi-

field?

Line 101: Were only eggs/larvae collected from the field? No pupae? Was water from the larval site brought for rearing?

Figure 2a,b. Are there statistical tests for UMAP allowing for the calculation of the significance of cluster separation?

Figure 2a,b: I would report somewhere---likely supplemental how balance each group size is. It seems like for *An. arabiensis/gambiae* there are far more samples from Tanzania (understandably for the regions they represent, but would be good to see the splits).

Fig 2e shows a broad sensitivity across wavelengths, including in areas of limited signal, i.e. a large peak in the flat spectral area around 1900. Assuming "sensitivity" here is a measure of feature (wavelength) importance towards prediction, do the authors have an explanation behind this? How far do you have to go (i.e. Supplemental Figure 1) to lose accuracy? Are all of the "main" models presented here using the full spectral region? This sensitivity is for which model, speciation or ageing?

Lines 146-151: I think here is where you need to establish the stability of predictive ability for each site through multiple independently reared data sets (I would imagine you have some of this data already based on your sample sizes).

Line 156: I'd argue that 17% here seems "relatively small" (line 143) only because your sample sizes are so large. 1200-1300 mosquito spectra is as large or larger than most of the datasets in the NIRS literature.

Line 172-182: Unless I'm missing something about transfer learning, it seems like the ability to retrain models successfully haven't been demonstrated through these datasets. Proof of success on independently reared data sets is the first step towards proof that wild-caught/semi-field reared larvae retrained models predict accurately wild-caught mosquitoes from that site.

Line 205-207: Along the same line as the above questions, what about mid-season stochasticity in samples. Temperature, rainfall, bacterial growth, etc. all could change over the course of a season.

Line 212: should be "to be generalised"

Lines 211-220: Similar issues with the framing. I don't believe you have shown here that this technique has the flexibility needed for wild mosquito prediction. It benefits to be clear about the potential challenges.

Line 217: I would love to see a robust assessment of wild, Polovodova graded mosquitoes as a training set. I think this is a great idea that gets around the wild/semi-field debate (though obviously is demanding and not without potential issues).

Lines 221-222: I wonder how much more "enhancement" might be feasible over what has already been done. This is a large sample size with 40,000 mosquitoes, and still it seems to fall into some of the same problems of smaller studies.

Supplementary Table 2: I would include (maybe not in this table, but otherwise) the metadata for these samples---sampling locations/dates/etc. I think this is important information to understanding the study and results as presented.

Supplementary Figures 2-5 don't seem to be referenced in the text. I would suggest discussion of them and inclusion of additional figures as suggested above.

The github link provided (<https://github.com/SimonAB/Mozzies-DL-MIRS-paper>) doesn't seem to work, though maybe it is still a private repository.

-Ben Krajacich - NIH Malaria Research Program Postdoctoral Fellow

Reviewer #2:

Remarks to the Author:

The impressive aspect of this manuscript is the sheer number of mosquitoes measured using FTIR spectroscopy. The sample size is indeed statistically significant and the results are indeed compelling but I have reservations about the sample preparation especially the amount of water that appears to differentiate the species. Figure 2C compares representative spectra from the three species under investigation. It is clear that one of the major differences is the water content between the three species. The band at $\sim 3500\text{ cm}^{-1}$ from the OH stretching mode of water is greater in *A. coluzii* compared to the other species. The range between $1000\text{--}500\text{ cm}^{-1}$ also shows clear water contributions as evinced by the increase in absorption in this range. Furthermore, the amide I region ($1700\text{--}1500\text{ cm}^{-1}$) also has a strong water contribution from the OH bending mode of water centred at 1635 cm^{-1} . It is clear that there is a major difference in the water content which has not been addressed and is possible a major contributor in the classification.

The authors have neglected to include other work investigating the mosquitoes using mid-infrared including a pivotal study by Khoshmanesh et al. *Anal Chem* 16;89(10):5285-5293. who applied ATR-FTIR to investigate age, sex and Wolbachia infection in *Aedes aegypti*. In that study the authors were able to deduce the age of mosquitoes and so this aspect of the study lacks novelty except for the fact it was done on different mosquitoes with shorter time points. They also reported that field studies resulted in a much lower sensitivity and specificity. This brings me to the main point that is not addressed. The mosquitoes are semi-field environment and therefore have similar blood meals and are grown in a semi-controlled environment. In order to have any real application the method must be tested on some real field samples.

The authors state that...

Training a CNN including mosquitoes reared in semi-field facilities from one country could not predict age 148 and species of populations from another (Study E2, Fig. 3). Similarly, training a 149 CNN including laboratory-reared mosquitoes from two sites could not predict age 150 and species of those reared at the third (Study E3, Fig. 4), even with pre-selected 151 wavenumbers (Study E4, Fig. 5).

So the chances of this working on a natural population seems remote given that it does not predict mosquitoes age from different countries.

Moreover, the species identification and age has also been reported by the authors in their earlier study, albeit using only two species and a much smaller sample cohort. Once again it is hard to see the novelty based on what has previously been published except for the impressive size of the study. The differences in the previous study need to be highlighted and what the is the novelty compared to the earlier study explained.

How can the authors be sure they are just recording spectra of the cuticle of a mosquito and not the internal structure?

I believe the authors are using ATR but that is not stated in the methods. For an ATR measurement the mosquito is placed onto the window and then a pressure clamp is applied, which would squash the mosquito hence how can the authors be sure that they are not picking up some of the internal chemistry of the mosquitoes?

According to Figure 3C one of the biggest regions of variation in mid-infrared absorption spectra of *An. arabiensis*, *An. coluzzii*, and *An. gambiae* form the three age classes is a band around 1900-1950 cm^{-1} but this is not mentioned in the text.

This concerns me as this is in a region devoid of absorbance and indicates the classification in part could be based on noise.

Given that one of the biggest variables for age would be size of the mosquito. Is the separation based on age due to a difference in total absorption or specific chemical differences. Was the data mean centred or normalised to take into account total absorbance?

Methodology

Critical pieces of information are missing in the spectroscopy e.g. Type of spectrometer, type of spectroscopy (ATR, FTIR transmission), number of scans for background and sample, methods on how the data was pre-processed.

Were second derivatives applied?

How was the baseline corrected?

What was the data pre-processing prior to modelling?

Minor point

By convention there is no such word as wavenumbers. This should be wavenumber values.

Response to reviewers

We are grateful to the two referees for the careful review of our original manuscript and for their excellent feedback that led to this significantly improved revised version. We are pleased that both referees valued the importance of this work and the interest in the extensive dataset produced. We have carefully considered their concerns about the general application of our method, the potential biases in the results due to background noise and issues with the methodology. In this revised version, we believe we have now satisfactorily addressed their comments and solved any potential issues relating to soundness of the methodology and reliability of the results. We have also vastly clarified the overall structure of the paper to emphasise its novelty and broader potential.

In particular, we have made two major changes to the manuscript: first, regarding the background noise and issues with the methodology, we have modified the presentation of the sensitivity analysis (see responses **R1.12** and **R2.8**) to allow an easier interpretation of the most important features in the mosquito spectra that are used by the model to make predictions on age and species (new Figure 4); we believe that it is now very clear that predictions are heavily based on regions that have very defined biochemical function. Secondly, regarding the general application of the methods, we have re-structured the results section to highlight the key innovation and advance of our approach based on the application of transfer learning to readily apply this technology to wild mosquitoes (see responses **R1.14** and **R2.4**). We believe that this crucial part of the analysis was not sufficiently clear in the previous version, leading to referees' concerns about the general application of the methods. In addition to these two major elements, 1) changed the sensitivity visualisation, and 2) the presentation of results, no other changes to models or results have been made.

Below is a detailed point-by-point response to the referees' suggestions. We are grateful for the careful review that our original manuscript received that led to this significantly improved revised version. We hope that with these revisions it will be now acceptable for publication in *Nature Communications*.

Reviewers' comments:

Reviewer #1 (Remarks to the Author):

R1.1 comment: *New methodology for the accurate and rapid age classification of mosquito disease vectors would be an enormous asset to the evaluation of anti-vector control measures and the field of vector biology as a whole. This paper "Rapid ageing and species identification of natural mosquitoes for malaria surveillance" by Siria et al. is an effort continuing their 2019 Wellcome Open Research manuscript on the use of mid-infrared spectroscopy (MIRS) to age/speciate Anopheles spp. mosquitoes. I commend the authors for their work in generating a very large dataset (almost certainly the largest of its kind), with lab/semi-field samples, multiple countries, species, and mosquito physiological statuses.*

The paper is well-written and scientifically sound description of their current efforts, though I worry that the tone is overly positive regarding the data as presented. A critical issue with past work in this

field using the related technique of Near Infrared Spectroscopy (NIRS) is that while you achieve good accuracy for laboratory colony mosquitoes, when you make the jump to field-caught material you essentially lose predictive power. This is likely due to a variety of genetic and environmental factors, which happily, the authors have attempted to account for here through their study design and the inclusion of mosquitoes from a variety of labs and rearing conditions. Unfortunately much like NIRS, it appears that MIRS may have similar issues with generalizability of the approach. The authors demonstrate that you must include some portion of mosquitoes from the test group (here 17%--- though with large sample sizes), for predictive ability to be achieved. I think that on its surface this necessary inclusion may be the “tax” necessary to use these techniques, but I feel that the authors have insufficiently demonstrated what this may look like in real world practice and the eventual move to field mosquitoes.

R1.1 response: We agree that recalibration of the model using target mosquitoes is a ‘tax’ warranted by this technique. However, we argue that this ‘tax’ is substantially lower than any currently available alternative. Indeed, the main advances and innovation of this work comes to the application of a transfer learning approach that minimises re-calibration efforts when moving to a new mosquito population. This is further explained in the response R1.14. We have also quantified what the ‘tax’ would be in the wild in association with measuring the impact of vector control efforts as presented in Figure 5. Finally, we have toned down overly positive presentation of the models.

R1.2 comment: *The study brings wild-caught larvae into semi-field environments to improve predictive ability, but it is not demonstrated that this inclusion gives you sufficient accuracy on new, independently reared groups of semi-field mosquitoes, nor (ideally) a selection of fully wild adult mosquitoes. While fully wild mosquitoes captured as adults would have unknown ages, proxies of age such as nulliparity (probably young), sporozoite positivity (certainly old), or even the general predicted age distribution of the population (skewed young) would be good indicators that the methodology is working. I acknowledge that analysis of full field adults is difficult, though this group certainly has the skills and ability to do so.*

R1.2 response: We agree that validating this technology on ‘fully’ wild mosquitoes using proxies of age would be an important future step in the implementation of this approach. However we believe this is out of the scope of this work considering the extensive amount of time and resources to conduct this proposed validation. Here, our aim required correctly aged-labelled mosquitoes, a ‘ground-truthing’ that is not possible with sufficient confidence in wild mosquitoes, as the reviewer acknowledges.

R1.3 comment: *If field mosquitoes are not available, I believe that at a minimum a demonstration of the accuracy of the technique on these different cohorts of semi-field reared mosquitoes would also be a sufficient demonstration that the technique is not overfitting and would have generalizable success. I think that transparency in these results is critical to pushing this avenue of research forward, and to not have other researchers purchase this instrument/putting time into this for wild mosquitoes erroneously believing it would give them success.*

R1.3 response: We believe we have demonstrated that using a transfer learning approach allows generalisable predictions, providing some examples of correctly-labelled mosquitoes are used for model ‘re-calibration’ for new populations. Please refer to response R1.14 for further details.

R1.4 comment: Line 30: Specific what this 95% accuracy refers to, species/age grading/both?--- comparing what model set to what test set?

R1.4 response: We agree that it is unclear and that it would be difficult to explain the different accuracies for the different datasets tested in this work in the abstract. For this reason we removed the accuracy and clarified that the models developed here predict both species and age simultaneously. It now reads: "Using over 40,000 ecologically and genetically diverse females, we could simultaneously speciate and age grade *An. gambiae*, *An. arabiensis*, and *An. coluzzii*".

R1.5 comment: Line 33-34: Final sentence of the abstract is awkward, maybe "In the future, we anticipate our method can be applied..."

R1.5 response: Modified as suggested.

R1.6 comment: Line 50, insert comma after "indistinguishable".

R1.6 response: Modified as suggested.

R1.7 comment: Line 82-86: are these "natural mosquito populations" or would it be more accurate to call them semi-field?

R1.7 response: We agree with this point and clarified in the text these mosquitoes were reared under semi-field conditions. It now reads: "In this study, we developed a MIRS approach to ultimately predict species and age of natural populations of three major African malaria vectors raised in semi-field mesocosms"

R1.8 comment: Line 101: Were only eggs/larvae collected from the field? No pupae? Was water from the larval site brought for rearing?

R1.8 response: Larvae, but not pupae, were collected from field sites in Tanzania and reared in the insectary using water from the field as already described in the methods: line 292: "Larvae were brought to the insectary and were sorted based on their morphology. The larvae were maintained in field water and provided with ground fish food (TetraMin®) until pupation." In Burkina Faso, blood fed and gravid mosquitoes were collected indoors and allowed to individually oviposit their eggs, then hatched larvae reared. This was also previously described in the methods: line 298: "all female mosquitoes, whether blood fed or gravid, were collected by trained technicians with mouth aspirators from local houses where mosquitoes have rested after a blood feeding. After aspiration the mosquitoes were transferred immediately into 30×30×30 cm cages covered with a wet cloth to avoid dehydration during transport. These mosquitoes were transferred to a room where light, humidity, and temperature are similar to that of the field (semi-field facility) and were maintained with glucose 5% for 72 hours to allow them to digest the blood. Then individual gravid females were transferred to a single cup containing 10ml water to allow oviposition."

R1.9 comment: Figure 2a,b. Are there statistical tests for UMAP allowing for the calculation of the significance of cluster separation?

R1.9 response: UMAP does not preserve inter-cluster distances between iterations because it includes a random seed, and those distances are very sensitive to the values chosen for the UMAP parameters; a statistical test of cluster separation would therefore be uninformative.

R1.10 comment: *Figure 2a,b: I would report somewhere---likely supplemental how balance each group size is. It seems like for An. arabiensis/gambiae there are far more samples from Tanzania (understandably for the regions they represent, but would be good to see the splits).*

R1.10 response: The details of the samples used in each of the presented models are reported in Supplementary Table 3 and Supplementary Table 4. However, we agree that we did not previously report information on the whole dataset, which we are now providing in Supplementary Table 2.

R1.11 comment: *Fig 2e shows a broad sensitivity across wavelengths, including in areas of limited signal, i.e. a large peak in the flat spectral area around 1900. Assuming "sensitivity" here is a measure of feature (wavelength) importance towards prediction, do the authors have an explanation behind this? How far do you have to go (i.e. Supplemental Figure 1) to lose accuracy? Are all of the "main" models presented here using the full spectral region? This sensitivity is for which model, speciation or ageing?*

R1.11 response: We agree that there were potential issues in the presentation of the sensitivity analysis of the model. We have inspected the method alongside considering newer analysis methods from the machine learning community to both fix and refine the sensitivity plots in the manuscript. While in our previous manuscript we represented the sensitivity of each normalised input feature (wavenumber values), we now represent the sensitivity in a more mechanistically-interpretable presentation allowing direct comparison across wavenumber values. Specifically, three adaptations led to the change of plots:

- 1) Considering each class (either species or age) separately highlights the sensitivities in the input spectra for each class, allowing easier comparison between sensitivities for classes, which may in turn provide interesting insight into differences between mosquito species as well as the aging of mosquitoes. Here only the true samples (i.e. spectra belonging to the specific class) for each class are used to create the sensitivities to that class.
- 2) Negative gradients are now zeroed, as used e.g. in the more recent GRAD-CAM papers (e.g.¹). This is because negative gradients represent wavenumbers that inhibit that class prediction and the interest here is to find wavenumbers that are corresponding positively to the model's classification. Negative gradients, corresponding to regions in the spectra that are detrimental to class prediction, will still be visible in the plots for the other classes (as positive contributions to them), so this information is not thrown away.
- 3) Whitening pre-processing is used on the input spectra and this had not been previously considered in the sensitivity analysis, which looked at the sensitivity to the network input. Therefore, the previous sensitivity analysis was with respect to the transformed spectra, massively expanding the low variance regions (essentially the low-intensity regions). We agree with the reviewers that that was confusing, so now we apply the inverse transformation of the pre-processing, such that the new sensitivity analysis is with respect to true input spectra, and is a much more easily interpretable visualisation, and gives much more intuitive results - we thank the reviewers for catching this issue. (we note that the network model itself is identical to the previous version - we only change the sensitivity visualisation).

R1.12 comment: *Lines 146-151: I think here is where you need to establish the stability of predictive ability for each site through multiple independently reared data sets (I would imagine you have some of this data already based on your sample sizes).*

R1.12 response: In the new version we presented the results on the predicted ability across different regions in lines 191-201. These results showed that despite the large quantity of data collected in this work, the lack of predictive accuracy across different labs and origins is not simply due to insufficient data but caused by slight differences between mosquito spectra in different regions. Upon demonstration that larger datasets would not be a solution, we then present a method using transfer learning, which we demonstrate provides a method for achieving high classification accuracy on mosquitoes from the semi-field. For this method to work, a small sample of mosquitoes from the target population is needed for updating the models; therefore, although the intermediate result is that there is little to no predictive ability for an independent dataset, there is a clear method for achieving strong predictive ability on an independent dataset that minimises sampling effort.

R1.13 comment: *Line 156: I'd argue that 17% here seems "relatively small" (line 143) only because your sample sizes are so large. 1200-1300 mosquito spectra is as large or larger than most of the datasets in the NIRS literature.*

R1.13 response: We agree that the term 'relatively' can be interpreted subjectively and might vary depending on the context, so we have removed it here.

R1.14 comment: *Line 172-182: Unless I'm missing something about transfer learning, it seems like the ability to retrain models successfully haven't been demonstrated through these datasets. Proof of success on independently reared data sets is the first step towards proof that wild-caught/semi-field reared larvae retrained models predict accurately wild-caught mosquitoes from that site.*

R1.14 response: In terms of transparency of the methods presented in paper, we have not tried to occlude the method in an attempt to present an "overly positive" tone, although we agree that the layout was not the best for providing clarity over the methods and results. We have now restructured the paper providing main results first and given more attention to the description of methods, specifically the transfer learning. The transfer learning approach is a standard approach in deep learning, when there is the availability of a large cheap to collect dataset that shares similarity with a more expensive smaller dataset that is of particular interest. Typical examples use pre-trained networks that have learned to classify a wide class of images (e.g ²), which means that they have many useful features for a new problem, but we need the data from that problem for the network to correctly associate these features with the target classes. These are now typically added to standard releases of software such as Tensorflow or PyTorch. We have adapted this approach to species identification and age grading of mosquitoes- using cheaply available lab data to build up the feature extraction capability, then re-calibrating on the realistic examples from a given domain.

The ability to re-train a model has been demonstrated on an independently reared dataset. The base model used in the transfer learning was trained on a large dataset of lab reared mosquitoes. For the transfer learning, we are then considering a dataset of semi-field reared mosquitoes. This dataset is completely independent from the lab dataset used during the initial training of the model. We then split this semi-field dataset into a training and testing set; perform the transfer learning using the training set; and test the ability for the re-trained with transfer learning model on its ability to predict semi-field reared mosquitoes from the testing set.

Under the assumption that semi-field reared mosquitoes are representative of field mosquitoes, this demonstrates that this method is very capable of predicting species and age of "wild" mosquitoes.

Further, this work presents the cost or “tax” associated with using these methods in the wild, in terms of how many mosquitoes need collecting for re-training of the model.

The presentation of results, mainly in the supplementary material, that show failings of the model only trained on lab data to predict mosquitoes from a different environment to those using in the training dataset, are presented to confirm results previously obtained using near-infrared spectroscopy³. The differences created by varying sources produce changes in MIRS to a sufficient level that models cannot then classify those mosquitoes from outside the origin of their training dataset. These results alone may give indication that it will be impossible for a model trained on lab data to work efficiently on wild mosquitoes in the field, but we present a transfer learning approach in this manuscript that overcomes this to demonstrate strong classification results on semi-field mosquitoes.

R1.15 comment: *Line 205-207: Along the same line as the above questions, what about mid-season stochasticity in samples. Temperature, rainfall, bacterial growth, etc. all could change over the course of a season.*

R1.15 response: This is certainly an interesting suggestion and will need to be addressed in future work. We have included this suggestion in the discussion in line 253.

R1.16 comment: *Line 212: should be "to be generalised"*

R1.16 response: We think that ‘to generalise’ is more appropriate in this context as it is the model that needs to generalise well to new mosquito populations, so we have not modified this sentence.

R1.17 comment: *Lines 211-220: Similar issues with the framing. I don't believe you have shown here that this technique has the flexibility needed for wild mosquito prediction. It benefits to be clear about the potential challenges.*

R1.17 comment: We believe we have now been more transparent on the ‘tax’ (re-calibration) needed for this approach to be implemented in the field. We have now focussed the results on the transfer learning approach which allows to generalise prediction minimizing re-calibration efforts. Please refer to R1.14 for additional details.

R1.18 comment: *Line 217: I would love to see a robust assessment of wild, Polovodova graded mosquitoes as a training set. I think this is a great idea that gets around the wild/semi-field debate (though obviously is demanding and not without potential issues).*

R1.18 response: As the reviewer rightly points out above, this is not without issues, especially when building a training set for which the ‘ground truth’, *i.e.*, correct labelling of samples, is paramount. Given the lack of accuracy of the Polovodova technique with regard to the age of the mosquitoes, wild mosquitoes for which date of emergence is unknown would only be useful as a validation set, and the accuracy of the model’s classification measured using extrinsic information (*e.g.*, concordance with Polovodova read-out; consistency with expectations for local mosquito population age structure; *etc.*). This is the reason we are using semi-field mosquitoes, which are the closest possible to wild mosquitoes except for their age being known.

R1.19 comment: *Lines 221-222: I wonder how much more “enhancement” might be feasible over what has already been done. This is a large sample size with 40,000 mosquitoes, and still it seems to fall into some of the same problems of smaller studies.*

R1.19 response: Please see our reply to the R1.14 comment.

R1.20 comment: *Supplementary Table 2: I would include (maybe not in this table, but otherwise) the metadata for these samples---sampling locations/dates/etc. I think this is important information to understanding the study and results as presented.*

R1.20 response: Information on the sampling location of field collected mosquitoes were described in the Methods section. We have now added the dates and species and it now reads (lines 259): “The experimental studies were conducted in two leading African malaria vector control institutions: Ifakara Health Institute (IHI), Tanzania and Institut de Recherche en Sciences de la Santé (IRSS), Burkina Faso. In IHI, the study experiments were conducted in the mosquito biology laboratory Vector Sphere and the larvae were collected from two villages in the Kilombero floodplains in Ulanga district, south-eastern Tanzania: Minepa village (longitude -8.285°, latitude 36.669°) and Tulizamoyo village (longitude -8.348°, latitude 36.732°) from April to July 2018. Morphologically identified *Anopheles gambiae* s.l. mosquitoes collected were almost uniquely *An. arabiensis*. The ecology and species available in Ulanga district were described by Kaindoa et al. In IRSS, mosquito sampling was conducted in the north of Bobo-Dioulasso in Vallée du Kou village (longitude -4.4201°, latitude 11.3824°) and in the south of Bobo-Dioulasso in Soumousso (longitude -4.0438°, and latitude 11.0125°) from July to October in 2017 and 2018. *Anopheles coluzzii* and *An. gambiae* were mainly present in Vallée du Kou and Soumousso, respectively.” We believe that the additional information presented in the Methods section together with information present in Supplementary Table 2 now provide all necessary information to understand the study design and samples included in the analysis.

R1.21 comment: *Supplementary Figures 2-5 don't seem to be referenced in the text. I would suggest discussion of them and inclusion of additional figures as suggested above.*

R1.21 response: All Supplementary figures are now referenced in the text.

R1.22 comment: *The github link provided (<https://github.com/SimonAB/Mozzies-DL-MIRS-paper>) doesn't seem to work, though maybe it is still a private repository.*

R1.22 response: The Github link will be made public upon publication. We have included the scripts as an attachment for review purposes.

Reviewer #2 (Remarks to the Author):

R2.1 comment: *The impressive aspect of this manuscript is the sheer number of mosquitoes measured using FTIR spectroscopy. The sample size is indeed statistically significant and the results are indeed compelling but I have reservations about the sample preparation especially the amount of water that appears to differentiate the species. Figure 2C compares representative spectra from the three species under investigation. It is clear that one of the major differences is the water content between the three species. The band at ~3500 cm⁻¹ from the OH stretching mode of water is greater in *A. coluzzii* compared to the other species. The range between 1000-500 cm⁻¹ also shows clear water contributions as evinced by the increase in absorption in this range. Furthermore, the amide I region (1700-1500 cm⁻¹) also has a strong water contribution from the OH bending mode of water centred at*

1635 cm^{-1} . It is clear that there is a major difference in the water content which has not been addressed and is possible a major contributor in the classification.

R2.1 response: The triangularly shaped band centred on 3300 cm^{-1} is not a bulk water band but caused by OH and NH stretches in the proteins and carbohydrates with contributions from water tightly bound to the proteins. The same is true for the other frequency ranges mentioned by the reviewer. We have shown the effects of not drying the mosquitoes in ⁴ and the relevant spectra are reproduced below. The mosquitoes used in this study were dried using a standardised protocol and therefore do not contain bulk water.

R2.2 comment: The authors have neglected to include other work investigating the mosquitoes using mid-infrared including a pivotal study by Khoshmanesh *et al.* *Anal Chem* 16;89(10):5285-5293. who applied ATR-FTIR to investigate age, sex and *Wolbachia* infection in *Aedes aegypti*. In that study the authors were able to deduce the age of mosquitoes and so this aspect of the study lacks novelty except for the fact it was done on different mosquitoes with shorter time points.

R2.2 response: We have now included a reference to Khoshmanesh *et al.* 2017 as well the recently published article by Srout *et al.* 2020, as previous studies that have applied MIRS on mosquitoes to classify age or species. However, we have made it clear that these studies used Partial Least Squares-Discriminant Analysis to analyse the spectra, which is a very different approach to what is presented in our work here. In addition to the type of analysis and species studied, our work specifically investigated the machine-learning approach on naturally reared mosquitoes, whereas the earlier studies solely analysed laboratory-reared mosquitoes. This aspect is one of the key novel points of our work. The text now reads (lines 80-84): “Additionally, MIRS was used to predict sex, age class (2 or 10 day old), and *Wolbachia* infection *Aedes aegypti* mosquitoes, as well classifying the species of *Aedes aegypti*, *Ae. albopictus*, *Ae. japonicus*, and *Ae. triseriatus* both by using Partial Least Squares-Discriminant Analysis on samples collected under laboratory-controlled conditions.”

R2.3 comment: *They also reported that field studies resulted in a much lower sensitivity and specificity. This brings me to the main point that is not addressed. The mosquitoes are semi-field environment and therefore have similar blood meals and are grown in a semi-controlled environment. In order to have any real application the method must be tested on some real field samples.*

R2.3 response: We agree that validating this technology on 'fully' wild mosquitoes, for example using proxies of age would be an important future step in the implementation of this approach. However we believe this is out of the scope of this work considering the extensive amount of time and resources to conduct this proposed validation. We also believe that semi-field mosquitoes do not grow in semi-controlled environment as the environmental parameters are virtually identical to the wild.

R2.4 comment: *The authors state that...*

Training a CNN including mosquitoes reared in semi-field facilities from one country could not predict age 148 and species of populations from another (Study E2, Fig. 3). Similarly, training a 149 CNN including laboratory-reared mosquitoes from two sites could not predict age 150 and species of those reared at the third (Study E3, Fig. 4), even with pre-selected 151 wavenumbers (Study E4, Fig. 5). So the chances of this working on a natural population seems remote given that it does not predict mosquitoes age from different countries.

R2.4 response: The results mentioned in this comment (now in lines 191-201) showed that despite the large quantity of data collected in this work, the lack of predictive accuracy across different labs and origins is not simply due to insufficient data but caused by slight differences between mosquito spectra in different regions. Upon demonstration that larger datasets would not be a solution, we presented a method based on transfer learning, which allowed us to achieve high classification accuracy on mosquitoes reared under semi-field conditions. The transfer learning approach is a standard approach in deep learning, when there is the availability of a large cheap to collect dataset that shares similarity with a more expensive smaller dataset that is of particular interest. Typical examples use pre-trained networks that have learned to classify a wide class of images (e.g ²), which means that they have many useful features for a new problem, but we need the data from that problem for the network to associate these features with the target classes. We have generalised this approach to the species identification and age grading of mosquitoes problem - using cheaply available lab data to build up the feature extraction capability, then re-calibrating on the realistic examples from a given domain. Therefore, although the intermediate result (lines 191-201) is that there is little to no predictive ability for an independent dataset, there is a clear method for achieving strong predictive ability on an independent dataset. Please see response to comment R1.14 for further information.

R2.5 comment: *Moreover, the species identification and age has also been reported by the authors in their earlier study, albeit using only two species and a much smaller sample cohort. Once again it is hard to see the novelty based on what has previously been published except for the impressive size of the study. The differences in the previous study need to be highlighted and what the is the novelty compared to the earlier study explained.*

R2.5 response: Please see above on response to comment R2.4. Briefly the novelty lies in generalisability using transfer learning as is now much more clearly explained in the manuscript. Thus, our work provides a method to get high classification accuracy on semi-field mosquito populations.

R2.6 comment: *How can the authors be sure they are just recording spectra of the cuticle of a mosquito and not the internal structure?*

R2.6 response: By using ATR, only the near-field region is probed, which is the mosquito cuticle and not much more. This was explained in more detail in ⁴.

R2.7 comment: *I believe the authors are using ATR but that is not stated in the methods. For an ATR measurement the mosquito is placed onto the window and then a pressure clamp is applied, which would squash the mosquito hence how can the authors be sure that they are not picking up some of the internal chemistry of the mosquitoes?*

R2.7 response: We acknowledge this has been omitted and we have now amended it in lines 329-332. Mosquitoes that had recently had a bloodmeal could cause dried blood to be squeezed out, which could affect the results. However, this has been taken care of as described in ⁴.

R2.8 comment: *According to Figure 3C one of the biggest regions of variation in mid-infrared absorption spectra of *An. arabiensis*, *An. coluzzii*, and *An. gambiae* form the three age classes is a band around 1900-1950 cm^{-1} but this is not mentioned in the text. This concerns me as this is in a region devoid of absorbance and indicates the classification in part could be based on noise.*

R2.8 response: We assume the reviewer is referring to figure 2(e) rather than 3(c) (and figure 4 in the new manuscript). We have changed the method for calculating the sensitivity of various spectral ranges to the final prediction of the model. While in our previous manuscript we represented the sensitivity of each normalised input feature (wavenumber values), we now represent the sensitivity in a more mechanistically-interpretable presentation allowing direct comparison across wavenumber values. This is also explained in more detail in our reply to R1.11 on figure 2(e). Briefly, we introduced three adaptations:

- 1) Considering each class (either species or age) separately highlights the sensitivities in the input spectra for each class, Here only the true samples (i.e. spectra belonging to the specific class) for each class are used to create the sensitivities to that class, thus allowing easier comparison between sensitivities for age or species.
- 2) Negative gradients are now zeroed, as these correspond to regions in the spectra that are detrimental to class prediction, so are less informative (but still be visible in the plots for the other classes).
- 3) The previous sensitivity analysis was with respect to the transformed spectra, massively expanding the low variance regions (essentially the low-intensity regions). As this was creating confusion, we now applied the inverse transformation of the pre-processing, such that the new sensitivity analysis is with respect to true input spectra, resulting in a much more easily interpretable visualisation (we note that the network model itself is identical to the previous version - we only change the sensitivity visualisation).

R2.9 comment: *Given that one of the biggest variables for age would be size of the mosquito. Is the separation based on age due to a difference in total absorption or specific chemical differences.*

R2.9 response: The body size of adult mosquitoes depends on larval development, but does not change over the adult lifespan, so variation in total absorption based on body size is possible but would not be associated with age, as age and adult body size are not correlated.

R2.10 comment: *Was the data mean centred of normalised to take into account total absorbance?*

R2.10 response: No, the spectra were neither mean centred nor normalised for analysis.

Methodology

R2.11 comment: *Critical pieces of information are missing in the spectroscopy e.g. Type of spectrometer, type of spectroscopy (ATR, FTIR transmission), number of scans for background and sample, methods on how the data was pre-processed.*

Were second derivatives applied?

How was the baseline corrected?

What was the date pre-processing prior to modelling?

R2.11 response: We have included the missing information in the spectroscopy in lines 329-332, it now reads: "The IR spectra of mosquitoes were acquired using Bruker Vertex 70 (UK) and Bruker ALPHA (Burkina Faso and Tanzania) FTIR spectrophotometers equipped with a diamond ATR accessory. Background and MIR spectra were acquired by averaging > 16 scans at a resolution of 4 cm⁻¹ over a range of 500–4,000 cm⁻¹. Mosquito spectra were cleaned and minor atmospheric intrusion compensated, while those with low intensity or a significant atmospheric intrusion were discarded automatically using a custom script as previously described.". We did not apply second derivatives (taking second derivatives is a technique for bringing out peaks in what would otherwise be quite smooth spectra. In our case, the ML approach can achieve the same effect by taking into account all the FTIR data) or correct the baseline in any way. The data were pre-processed prior to modelling as described in ⁴ and previously mentioned in the manuscript. Finally, the baseline was corrected using the StandardScaler package from scikit learn to pre-process all the spectra (all scripts will be made public upon publication in the Github repository provided).

Minor point

R2.11 comment: *By convention there is no such word as wavenumbers. This should be wavenumber values.*

R2.11 response: Corrected as suggested.

Reviewers' Comments:

Reviewer #1:

Remarks to the Author:

The manuscript of Siria et al. on the use of MIRS for the age-grading of Anopheles mosquitoes has been improved by the response to both the reviewers, and overall I feel that the paper is well analyzed and presented with the technique showing promise. However I still feel the fundamental gap in the paper remains, and the authors have not adequately proved that the transfer learning technique will be successful in adapting the approach to the field, nor stated what the sampling for this technique would entail (i.e. do you need a semi-field Vector Sphere?). Novelty in the realm of mosquito aging is proof on true wild specimens, not just on your lab or semi-field samples.

I fully appreciate the work required for the field validation of these approaches, and do think the simulation studies presented are useful to getting to this validation, but I don't think a base demonstration of the approach is outside the scope of this paper. As I mentioned a base example of the technique would be to just do a random aspiration sampling of adult mosquitoes in an area you have applied the transfer learning technique on the ~300 mosquitoes mentioned. If the age distribution looks plausible (all the better if you get some sporozoite positive old mosquitoes, which shouldn't be too difficult with a few hundred mosquitoes), then you have provided much more convincing evidence than the simulation study alone. When you present a dataset of 40,000 mosquitoes from two laboratories with ready access to field material, I think this isn't outside the capabilities of the authors. Additionally, as you present this working on dried mosquitoes, certainly someone must have dried adult mosquitoes available from these villages, I know that both Vallee du Kou and Soumouso are heavily sampled.

I also think that the R1.12 response about independently reared datasets misses the mark, there is room here to say something to the extent of "after transfer learning using June 2019 Soumouso-origin vector sphere mosquitoes, we were able to predict July 2019 Soumouso-origin vector sphere mosquitoes" or something like that. I understand between two different labs the technique doesn't work without transfer learning, but for the field researcher trying to evaluate their vector control measure, it is unclear how MIRS could be used and what transfer learning datasets are required.

I don't think the paper should be rejected in its current state, though I fear readers may misinterpret the results that this approach certainly will work on wild mosquitoes. I do not believe this has been shown, and statements like "These results demonstrate how this low-cost, artificial intelligence-based approach can quantify previously immeasurable impacts of interventions on natural vector populations, and constitute a new surveillance tool in the fight against malaria" in lines 95-98 could push the reader in this direction.

Reviewer #2:

Remarks to the Author:

The authors have addressed most criticisms adequately and included some of the necessary spectroscopic details required. They have also made the case for the spectra being devoid of water. There are some points to address.

1. It is not clear how many replicate spectra were taken for each mosquito in the in the "Spectroscopy" section. Were spectral replicates used in the model?
Does the test set include any replicate spectra exposed in the model?

2. The authors split the data set into a 10 % stratified test set and 90 % model. What happens if the data set is split into a 1/3 and the model 2/3?

This is a more standard way to split a model and test set especially when there is such a large data

set to model on. This would give more of an indication of the robustness of the model.
In fact a much more robust way to model data is to use double cross-validation and have randomised test and model sets.
Stratification does not automatically reduce bias in the modelling and should also be varied.
<https://doi.org/10.1186/1758-2946-6-10>

3. How many mosquito spectra were removed from the model because of the low absorbance?
This needs to be specified.

4. Was there any pressure applied to the mosquito to achieve maximum contact with the internal refraction element?
This needs to be spelt out in the methodology.

5. What do the authors mean by the spectra were "cleaned"?
Does this mean smoothed or removed? If smoothed what type of smoothing and how many smoothing points? If removed then see point 3.

6. There is still the word "wavenumbers" used in the manuscript instead of wavenumber values.

Response to reviewers

We are grateful to the two referees for the careful review of our revised manuscript and for their crucial feedback that led to what we believe are substantial improvements in this revised version. We are pleased that both referees valued the previous revision and highlighted the potential high impact and relevance of the work. The key remaining issue flagged in our previous revision was the need to more robustly demonstrate that the transfer learning technique we incorporated can be successfully applied to wild mosquitoes. We have put extensive efforts into addressing this over the last year through initiation of further data collection and analysis to generate the presented findings. The result is what we see as a convincing and unambiguous demonstration that this approach can be applied to naturally, wild collected mosquitoes. In particular, we highlight that these results demonstrated that our approach can estimate the age of wild malaria vectors encompassing two different species and two study sites in East and West Africa. We believe this represents a major breakthrough that will lead to significant improvements in the surveillance and control of mosquito vector-borne diseases.

Additionally, our revision addresses other minors points as detailed in the point-by-point response to the referees' below. We hope that with these revisions it will be now acceptable for publication in *Nature Communications*.

Our code, minimum working examples, and relevant instructions can be consulted in the README file of our GitHub repository:

https://github.com/SimonAB/DL-MIRS_Siria_et_al/blob/master/README.md

Reviewers' comments:

Reviewer #1 (Remarks to the Author):

R1.1. comment: *The manuscript of Siria et al. on the use of MIRS for the age-grading of Anopheles mosquitoes has been improved by the response to both the reviewers, and overall I feel that the paper is well analyzed and presented with the technique showing promise. However I still feel the fundamental gap in the paper remains, and the authors have not adequately proved that the transfer learning technique will be successful in adapting the approach to the field, nor stated what the sampling for this technique would entail (i.e. do you need a semi-field Vector Sphere?). Novelty in the realm of mosquito aging is proof on true wild specimens, not just on your lab or semi-field samples. I fully appreciate the work required for the field validation of these approaches, and do think the simulation studies presented are useful to getting to this validation, but I don't think a base demonstration of the approach is outside the scope of this paper. As I mentioned a base example of the technique would be to just do a random aspiration sampling of adult mosquitoes in an area you have applied the transfer learning technique on the ~300 mosquitoes mentioned. If the age distribution looks plausible (all the better if you get some sporozoite positive old mosquitoes, which shouldn't be too difficult with a few hundred mosquitoes), then you have provided much more convincing evidence than the simulation study alone. When you present a dataset of 40,000 mosquitoes from two laboratories with ready access to field material, I think this isn't outside the capabilities of the authors. Additionally, as you present this working on dried mosquitoes, certainly someone must have dried adult*

mosquitoes available from these villages, I know that both Vallee du Kou and Soumouosso are heavily sampled.

R1.1. response: We have carefully considered this suggestion and agreed on the substantial added value from including assessment of the performance of the DL-MIRS approach relative the gold standard technique for ageing wild malaria vectors, e.g. characterisation of the number of gonotrophic cycles that females have undergone as assessed via ovarian dissection. We initiated further field collections of wild malaria vectors from villages in Tanzania and Burkina Faso to generate additional data for comparison of these methods (full details in lines 225-255; 367-392; 447-457; 479-484). We found that utilising only 335 and 758 mosquitoes for the transfer learning in Burkina Faso and Tanzania, respectively, our DL-MIRS approach predicted very similar age structures to those obtained from ovarian characterization (Figure 6), suggesting our method is highly concordant with the existing gold stand and thus suitable for implementation in the field. The ovarian characterization method does have its own limitations (as described in lines 276-279), however it is the best available independent metric of the age of mosquitoes in wild population. Together with the strong performance of the DL-MIRS approach in predicting mosquitoes of known age in lab and semi-field settings, we believe this provides convincing evidence that DL-MIRS can be used for age-grading of wild mosquitoes.

R1.2. comment: *I also think that the R1.12 response about independently reared datasets misses the mark, there is room here to say something to the extent of "after transfer learning using June 2019 Soumouosso-origin vector sphere mosquitoes, we were able to predict July 2019 Soumouosso-origin vector sphere mosquitoes" or something like that. I understand between two different labs the technique doesn't work without transfer learning, but for the field researcher trying to evaluate their vector control measure, it is unclear how MIRS could be used and what transfer learning datasets are required.*

R1.2. response: We agree with the reviewer that a clear description on how DL-MIRS should be implemented in the field should be indicated, including in the context of the evaluation of vector control measures. Indeed, we believe we have clearly provided this information in the manuscript, where we showed evidence from semi-field experiments (Figure 3) combined with simulations (Figure 5) to understand the sample size needed for both 1) the transfer learning and 2) the target populations to evaluate two hypothetical vector control strategies (with high or low killing effects). Specifically, in Figure 5 we are showing the number of target populations required by DL-MIRS to detect an age structure shift (expressed as statistical power) based on the accuracy of the model, which depends on the amount of transfer learning, for example 324 semi-field spectra give >80% accuracy. As described in the methods (and Supplementary Figure 3), in the transfer learning process these spectra were balanced between the three age classes, each containing 108 spectra. In this revision, we have used 335 and 758 spectra of wild mosquitoes for the transfer learning of target populations in Burkina Faso and Tanzania, respectively; notably, these datasets were not balanced between the three age classes (the older class included only 26 and 17 spectra in each site) and thus are potentially less informative for the transfer learning; however, even with this limitation DL-MIRS showed similar age structures compared to ovarian characterisation, further suggesting that the details presented in Figure 5 and Supplementary Table 5 provide appropriate indications on how DL-MIRS should be implemented in the field.

R1.3 comment: I don't think the paper should be rejected in its current state, though I fear readers may misinterpret the results that this approach certainly will work on wild mosquitoes. I do not believe this has been shown, and statements like "These results demonstrate how this low-cost, artificial intelligence-based approach can quantify previously immeasurable impacts of interventions on natural vector populations, and constitute a new surveillance tool in the fight against malaria" in lines 95-98 could push the reader in this direction.

R1.3. response: We believe we now have addressed this point through much inclusion of much more convincing evidence that this method can predict the age structure of wild mosquito populations based on DL-MIRS (Figure 6). However, we have also modified the text according to the suggestion. This now reads: "These results demonstrate how this low-cost, artificial intelligence-based approach can determine the age structure of natural vector populations, and constitute a new surveillance tool in the fight against malaria."

Reviewer #2 (Remarks to the Author):

The authors have addressed most criticisms adequately and included some of the necessary spectroscopic details required. They have also made the case for the spectra being devoid of water. There are some points to address.

R2.1 comment: *It is not clear how many replicate spectra were taken for each mosquito in the in the "Spectroscopy" section. Were spectral replicates used in the model? Does the test set include any replicate spectra exposed in the model?*

R2.1 response: A single spectrum was taken from each mosquito. This was decided because normally the surface of the ATR glass is larger than the mosquito itself and because using the anvil of the ATR to collect spectra severely damages the cuticle of the mosquitoes. We have added a sentence to the experimental section to clarify this point.

R2.2 comment: *The authors split the data set into a 10 % stratified test set and 90 % model. What happens if the data set is split into a 1/3 and the model 2/3? This is a more standard way to split a model and test set especially when there is such a large data set to model on. This would give more of an indication of the robustness of the model. In fact a much more robust way to model data is to use double cross-validation and have randomised test and model sets. Stratification does not automatically reduce bias in the modelling and should also be varied. <https://doi.org/10.1186/1758-2946-6-10>*

R2.2 response: We thank the reviewer for the suggestion to use an alternative dataset splitting method. The proportions and sources of the training and unseen test samples vary depending on the models and tests of generalisability as described in the main text. The cross-validation scheme used for the CNN within the training set was 10-fold and randomised with a fixed random seed (in Keras: `KFold(n_splits=10, shuffle=True, seed=16)`). Using a splitting method that makes less data available to the model for training such as the one suggested by the reviewer would only really test how well the model learns with less data. We decided to use a 10-fold splitting scheme as 10% validation sets were large enough to cover a wide range of examples across the dataset while maximising the training set, allowing us to capture as much useful variation in the MIRS models as possible.

R2.3 comment: *How many mosquito spectra were removed from the model because of the low absorbance? This needs to be specified.*

R2.3 response: In the LV dataset 1416 out of a total of 29239 spectra were removed due to atmospheric intrusion, high water content or low intensity; similarly, in the GV dataset 640 out of 10258 were discarded, while in the EV and wild datasets 244 (out of 3521) and 11 (out of 1104) spectra were removed, respectively. We have now added this information in Supplementary Table 2.

R2.4 comment: *Was there any pressure applied to the mosquito to achieve maximum contact with the internal refraction element? This needs to be spelt out in the methodology.*

R2.4 response: Yes, to maximise contact between the mosquito cuticle and the surface of the ATR crystal, the anvil of the instrument was used. Thank you for this observation, we have added this detail to the experimental section.

R2.5 comment: *What do the authors mean by the spectra were "cleaned"? Does this mean smoothed or removed? If smoothed what type of smoothing and how many smoothing points? If removed then see point 3.*

R2.5 response: The spectra were never altered in any way, not even by smoothing them out. What we mean by "cleaned" is that the spectra taken that were not of sufficient quality were not used in the analysis, in other words, they were removed. We have clarified this in the text (in lines 401-403), where we stated that "Mosquito spectra were cleaned and minor atmospheric intrusion compensated, while those with low intensity or a significant atmospheric intrusion were discarded automatically using a custom script as previously described."

R2.6 comment: *There is still the word "wavenumbers" used in the manuscript instead of wavenumber values.*

R2.6 response: We have now removed the word wavenumbers and replaced it with wavenumber values (or wavenumber bands) through the text.

Reviewers' Comments:

Reviewer #1:

Remarks to the Author:

This paper, "Rapid age-grading and species identification of natural mosquitoes for malaria surveillance", has undergone significant revisions from my initial review. First, I am pleased the authors have taken the suggestion to expand the scope of this paper into the critical and important usage of this technique on wild vectors. I believe the paper has been greatly strengthened by the inclusion of this aspect, and is fit for publication.

Focusing primarily on the inclusion of the wild mosquitoes, as this is the large change in this version of the manuscript, I have a few minor suggestions.

In Supplemental Table 6, I would include a total row per village to better understand numbers of each condition in relation to the unbalanced nature of the test set. It would also be helpful to understand what percentage are included for transfer learning and how this may cause (or not) overfitting. I know some of this is in Supplemental table 3, but I'm not seeing how the groups were chosen.

Why is Figure 6 a bar chart rather than the confusion matrix used for most of the other figures? I think both have merits, just understanding if this changes the interpretation.

I didn't see a cost estimate for the MIRS machine, and any discussion of how suited to limited power/field situations could be valuable additions.

Response to reviewer

We are grateful to the referee for the careful review of our revised manuscript and for their additional feedback. We are pleased that they valued this revision and considered that the paper has been greatly strengthened by including the application of this technique on wild malaria vectors.

We have now addressed the final minor points as detailed in the point-by-point response to the referee below.

Reviewer #1 (Remarks to the Author):

R1.1. comment: *In Supplemental Table 6, I would include a total row per village to better understand numbers of each condition in relation to the unbalanced nature of the test set.*

R1.1. response: We agree with the reviewer, and we added a row per village that summaries the total number of samples in each group.

R1.2. comment: *It would also be helpful to understand what percentage are included for transfer learning and how this may cause (or not) overfitting. I know some of this is in Supplemental table 3, but I'm not seeing how the groups were chosen.*

R1.2. response: This information was already included in Supplementary Table 3, however we have made this more explicit and indicated also the percentage of wild data used for the transfer learning as requested. The section explaining the training and test sets for the wild mosquito model in Supplementary Table 3 now reads:

“Implicit use of Data from groups LV and (7200 data points). Data balanced by country, species, and age groups for groups LV and GV, and unbalanced for the wild group. In Burkina Faso, the transfer learning with the wild data set is composed of 205 G0, 104 G1, 26 G234 (total 335 data points, 4.4% of the whole training set). In Tanzania, the transfer learning with the wild data set is composed of 168 G0, 573 G1, 17 G234 (total 758 data points, 9.5% of the whole training set). Testing data set is wild data set of 568 (Burkina Faso) and 834 (Tanzania) non-dissected mosquito data points not included in the training set.”

R1.3. comment: *Why is Figure 6 a barchart rather than the confusion matrix used for most of the other figures? I think both have merits, just understanding if this changes the interpretation.*

R1.3. response: We could not use a confusion matrix to present the results on wild mosquitoes. As a confusion matrix represents instances in the actual class and instances in the predicted class, it implicitly needs the true class to be known. Therefore, to validate our approach, in Figure 6 we compared the proportion of predicted age classes of non-dissected wild mosquitoes (whose true age class is unknown) with the expected proportion of age classes based on dissected wild mosquitoes (which enabled us to classify the mosquito age class). We believe we have extensively explained our approach in the methods, results and Figure 6 caption, for example in the result section we write: “The number of gonotrophic cycles passed was morphologically identified (Supplementary Table 6). After dissection, mosquitoes were dried and scanned by MIRS. Gonotrophic cycle classification was used to (i) estimate the overall physiological age structure over the collection period at each site, and (ii) provide known age classes for transfer learning of DL-MIRS spectra. To independently test model predictions, we also scanned non-dissected mosquitoes, selected at random from the same populations as the dissected ones. Here, we assumed that the age structure of dissected and non-

dissected mosquitoes should be similar. [...] DL-MIRS predicted very similar age structures for non-dissected (test) and dissected (morphologically assessed) wild mosquitoes (Fig. 6).”

R1.4. comment: *I didn't see a cost estimate for the MIRS machine, and any discussion of how suited to limited power/field situations could be valuable additions.*

R1.4. response: Our method requires the purchase of an ATR-FTIR spectrometer costing around \$20,000. This is certainly a large initial investment, but the savings in operator time and consumables make it a competitive method, in the long run, especially considering that most manufacturers guarantee 8-10 years of continuous use. We added a sentence in the conclusions stating the cost estimate for the MIRS machine. This reads: “This approach would require an initial investment for the ATR-FTIR spectrometer (~\$20,000), but no other costs will be virtually needed in the long-term.